# Genome-wide association studies reveal the role of polymorphisms affecting factor H binding protein expression in host invasion by *Neisseria meningitidis*

Sarah G. Earle[1,☯], Mariya Lobanovska[2☯], Hayley Lavender[2☯], Changyan Tang[3], Rachel M. Exley[2], Elisa Ramos-Sevillano[2], Douglas F. Browning[4¤], Vasiliki Kostiou[5], Odile B. Harrison[6], Holly B. Bratcher[6], Gabriele Varani[3]*, Christoph M. Tang[2]*, Daniel J. Wilson[1,7]*, Martin C. J. Maiden[6]*

**1** Big Data Institute, Nuffield Department of Population Health, University of Oxford, Li Ka Shing Centre for Health Information and Discovery, Oxford, United Kingdom, **2** Sir William Dunn School of Pathology, University of Oxford, Oxford, United Kingdom, **3** Department of Chemistry, University of Washington, Seattle, Washington United States of America, **4** School of Biosciences, University of Birmingham, Edgbaston, Birmingham, United Kingdom, **5** Nuffield Department of Clinical Medicine, Experimental Medicine Division, John Radcliffe Hospital, Oxford, United Kingdom, **6** Department of Zoology, University of Oxford, Oxford, United Kingdom, **7** Department for Continuing Education, University of Oxford, Oxford, United Kingdom

☯ These authors contributed equally to this work.
¤ Current address: College of Health & Life Sciences, Aston University, Birmingham, United Kingdom
* varani@uw.edu (GV); christoph.tang@path.ox.ac.uk (CMT); daniel.wilson@bdi.ox.ac.uk (DJW); martin.maiden@zoo.ox.ac.uk (MCJM)

**Data Availability Statement:** All data are available within the manuscript at its supporting files. Sequence data are available through PubMLST.org/

## Abstract

Many invasive bacterial diseases are caused by organisms that are ordinarily harmless components of the human microbiome. Effective interventions against these microbes require an understanding of the processes whereby symbiotic or commensal relationships transition into pathology. Here, we describe bacterial genome-wide association studies (GWAS) of *Neisseria meningitidis*, a common commensal of the human respiratory tract that is nevertheless a leading cause of meningitis and sepsis. An initial GWAS discovered bacterial genetic variants, including single nucleotide polymorphisms (SNPs), associated with invasive meningococcal disease (IMD) *versus* carriage in several loci across the meningococcal genome, encoding antigens and other extracellular components, confirming the polygenic nature of the invasive phenotype. In particular, there was a significant peak of association around the *fHbp* locus, encoding factor H binding protein (fHbp), which promotes bacterial immune evasion of human complement by recruiting complement factor H (CFH) to the meningococcal surface. The association around *fHbp* with IMD was confirmed by a validation GWAS, and we found that the SNPs identified in the validation affected the 5' region of *fHbp* mRNA, altering secondary RNA structures, thereby increasing fHbp expression and enhancing bacterial escape from complement-mediated killing. This finding is consistent with the known link between complement deficiencies and CFH variation with human susceptibility to IMD. These observations demonstrate the importance of human and bacterial genetic variation across the fHbp:CFH interface in determining IMD susceptibility, the transition from carriage to disease.

neisseria as described in the manuscript and its supporting files.

**Funding:** Work in MCJM's Laboratory in Oxford is funded by the Wellcome Trust (https://wellcome.org/) grant numbers 087622/Z/08/Z and 218205/Z/19/Z, and the Meningitis Research Foundation (https://www.meningitis.org/, grant number 1901). Work in CMT's laboratory is funded by the Wellcome Trust (https://wellcome.org/, grant number 102908/Z/13/Z). DJW is a Sir Henry Dale Fellow, jointly funded by the Wellcome Trust and the Royal Society (grant number 101237/Z/13/B) and is supported by a Big Data Institute Robertson Fellowship. Computation using the Oxford Biomedical Research Computing (BMRC) facility is supported by Health Data Research UK and the NIHR Oxford Biomedical Research Centre (https://oxfordbrc.nihr.ac.uk/). Financial support was provided by a Wellcome Trust (https://wellcome.org) Core Award, grant number 203141/Z/16/Z. Work by GV at the University of Washington was supported by National Institute of Health, National Institute of General Medical Sciences (NIGMS, https://www.nigms.nih.gov/) grant number, 1R35 GM 126942. SGE, ML, HL, RME, ER-S, VK, OBH, and DJW received salary support from the Wellcome Trust. HBB received salary support from the Meningitis Research Foundation. CT and GV received salary support from National Institute of General Medical Sciences. The funders had no role in the study design, data collection and analysis, decision to publish or preparation of the manuscript.

**Competing interests:** GV is a co-founder of Ithax Pharmaceuticals and Ranar Therapeutics. All other authors have declared that no competing interests exist.

## Author summary

The human bacterial pathogen *Neisseria meningitidis* (the meningococcus) causes sepsis and meningitis worldwide. Paradoxically, it is carried harmlessly in the nose and throat far more commonly than it causes invasive disease, raising the question of whether the bacteria causing disease are genetically the same as those that do not. We looked for systematic differences in the DNA of the bacteria from invasive disease and those that were carried harmlessly. Whilst controlling for population structure as a confounder, we identified multiple genes apparently contributing to invasion and identified a signal around the gene encoding an important component of some meningococcal vaccines, called factor H binding protein (fHbp). This protein helps the meningococcus evade killing by the human immune response by binding to a human defence protein called complement factor H. We validated this result in an independent population of meningococci and identified that DNA variants that control the amount of fHbp produced by the bacteria were important, with high levels of fHbp expression associated with invasion. Given that human DNA variation in complement factor H increases the risk of meningococcal invasion, this highlights the important connection between human and bacterial genetic variation and helps explain why some people infected with meningococci suffer meningococcal disease whilst others do not.

## Introduction

Many normally harmless members of the human microbiota can cause invasive disease in certain circumstances. Given the complexity of relationships between commensal and symbiotic bacteria and their hosts, there are numerous factors that promote the disruption of asymptomatic interactions, resulting in host tissue invasion, including genetic polymorphisms and phenotypic changes in hosts and infecting microbes. These diseases present both an evolutionary puzzle, as host invasion is often a dead-end for transmission, and an epidemiological challenge, as the aetiological agent circulates widely undetected, striking seemingly at random. An important example of such an 'accidental' pathogen is *Neisseria meningitidis*, the meningococcus, a common coloniser of the human oropharynx [1,2], which occasionally causes devastating invasive meningococcal disease (IMD).

Human populations are affected by IMD globally, with incidence that varies widely by time, geographical region, and host age [3]. This reflects the high variability of the meningococcus both genetically and antigenically [4], with many variants arising because of frequent intraspecies and occasional interspecies horizontal genetic exchange (HGT) [5]. The resultant genetic variation is structured by clonal complex (cc), which can be determined by MLST [6], and is a surrogate for genetic lineage [4]. The ccs tend to be associated with particular antigen variants, especially the polysaccharide capsule, the meningococcal serogroup antigen. Importantly, although the majority of meningococci have low invasive potential, certain cc:serogroup combinations, known as 'hyperinvasive lineages', are associated with IMD, often with distinct clinical or epidemiological manifestations [7]. Therefore, as hyperinvasive meningococci spread through populations in asymptomatic carriage, the nature and incidence of IMD changes [3,8].

The molecular mechanisms involved in the transition of asymptomatic colonisation to IMD remain poorly defined despite extensive research. Whilst invasive meningococci almost invariably express one of the disease-associated capsular polysaccharides (corresponding to serogroups A, B, C, W, X, or Y), and host complement deficiencies are a well-known risk

factor [9], these two risk factors cannot explain the variation in invasiveness observed among different hosts or bacterial variants. Host susceptibility to IMD is also linked to genetic variation in the locus encoding the negative regulator of the complement system, complement factor H (CFH) and CFH-related protein 3 (CFHR3); both these host proteins bind to meningococcal fHbp, an important vaccine antigen [10]. High affinity binding of CFH to fHbp promotes *N. meningitidis* evasion of the complement system and survival in serum and this can be countered by competition with CFHR3 [10–14].

Several other meningococcal cell components have been associated with invasion including sub-capsular surface antigens, the meningococcal disease associated island (MDA), pili, toxins, and iron transporters [15]; however, all these components are also distributed among less pathogenic meningococci and other *Neisseria* species that are not regularly associated with invasive disease [16,17]. Therefore, the meningococcal invasive phenotype, which can be measured by the relative prevalence of meningococcal genomes in asymptomatic carriage and disease [18], is polygenic and appears to be different for different invasive meningococci [8].

We leveraged the diversity of meningococcal genotypes, antigen types, and invasive potential to investigate the meningococcal invasive phenotype with a hypothesis-free approach using two genome wide association studies (GWASs) of isolates from invasive disease and asymptomatic carriage. Importantly, we controlled for population structure, which can confound results. This identified meningococcal genetic polymorphisms throughout the genome, highlighting many genes known to be associated with invasive potential. One variant affected the expression of fHbp, with higher levels of expression associated with serum resistance. Our findings are consistent with an independent investigation, published after our study was completed, which used a candidate gene approach to identify upstream variants of *fHbp* that affected fHbp expression and meningococcal survival in human serum. Taken together these two independent studies confirm that intergenic regions upstream of *fHbp* that cause higher expression of fHbp, an important vaccine antigen, are associated with increased risk of IMD in humans [19].

## Results

### GWASs

We explored the relationship between meningococcal genetic variation and IMD in two genome-wide association studies (GWAS) encompassing 1,556 genomes. Initially, we compared *N. meningitidis* isolates from a well-characterised set of 52 cases of IMD and 209 carriers collected in the Czech Republic [20–22], in which a preponderance of disease isolates belonged to the hyperinvasive ST-11 clonal complex (cc11; O.R. 3.4, 95% CI 1.7–7.1, Wald test $p = 10^{-3.90}$) [6,23] (Figs 1A and S1). Heritability was substantial, with 36.5% (95% CI 15.9–57.0%) of the variability in case-control status attributable to bacterial genotype. GWAS identified 17 loci harbouring variants associated with carriage *versus* disease, after controlling for strain-to-strain differences using a linear mixed model [24]. In total, we tested for associations at 156,804 SNPs mapped to a cc11 reference genome [25] and 7,806,583 31-nucleotide sequence fragments (kmers) that capture variation missed by reference-based SNP calling [26]. After Bonferroni correction for unique phylopatterns [27], we found significant associations in seven SNPs ($p<10^{-6.20}$) and 465 kmers ($p<10^{-6.79}$) across the genome (Fig 1B and 1C). This identified variation in several genes associated with known virulence factors including the *cps* region, encoding the capsule, and the phage-encoding meningococcal disease-associated (MDA) island (Fig 1D–1F) [28–31].

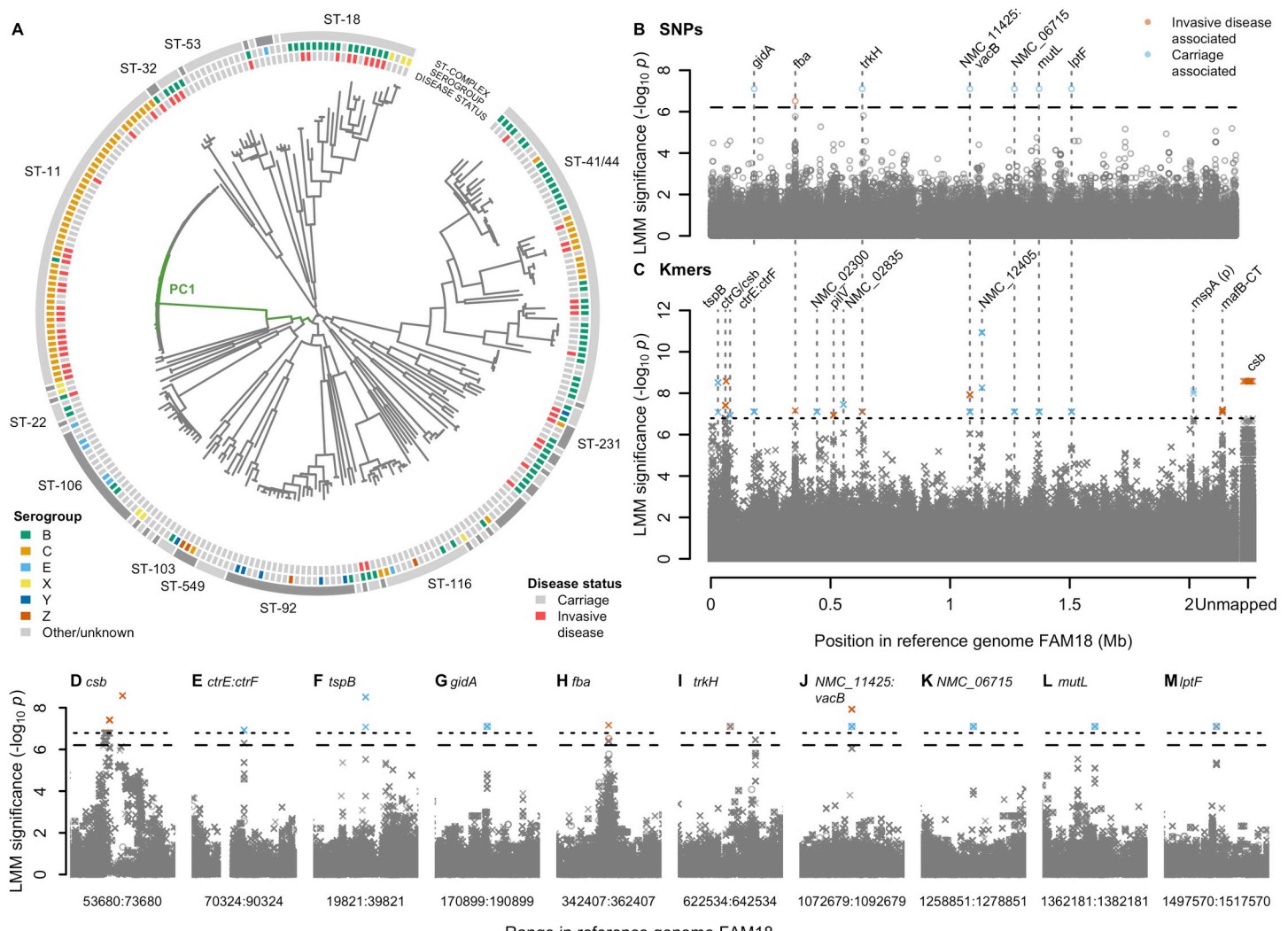

**Fig 1.** **(A)** Phylogeny of 261 *N. meningitidis* strains sampled from the Czech Republic in 1993 shows a strong strain association between invasive disease and the ST-11 complex. Clonal complexes are shown in the outer grey ring. Serogroups are shown on the next ring inwards. Disease status is shown on the next ring, invasive disease (red, n = 209) or carriage (grey, n = 52). Branches of the phylogeny most correlated with the significantly associated PC 1 are coloured in green. (B-J) SNPs and kmers associated with carriage *vs*. invasive disease in the 261 isolates. Significant SNPs and kmers are coloured by the LMM estimated direction of effect. Bonferroni-corrected significance thresholds are shown by black dashed (SNPs) and dotted (kmers) lines. Gene names separated by colons indicate intergenic regions. FAM18 reference genome gene name prefixes have been shortened from NMC_RS to NMC_. **(B)** Each point represents a SNP aligned to the reference genome FAM18. **(C)** Each point represents the left-most position of a kmer in the reference genome FAM18 based on mapping and BLAST alignments. Unmapped kmers are plotted to the right of the Manhattan plot. (D-M) Close ups of genes containing significant SNPs (circles) or kmers (crosses) +/- 10kb, SNPs and kmers are shown.

## Phase variable region in *csb*

A total of 21 kmers in the serogroup B encoding gene *csb* (formerly *siaD_B* and *synD* [32]), encoding the capsule polysialyltransferase, were associated with elevated disease risk ($p = 10^{-8.58}$) and mapped to a poly-A tract in the coding region which mediates ON:OFF switching of capsule expression (S2 Fig) [33,34]. The significant kmers, present in 68 isolates, all contained poly-A tract lengths of 9 and were bioinformatically predicted to be representing the serogroup B *csb* "on" state within the poly(A) phase variable region, determined by whether the assembled gene contained frameshifts or premature stop codons relative to the *N. meningitidis* isolate MC58.

## Putative promoter in the intergenic region between *ctrE* and *ctrF*

Nine kmers that tagged the consensus haplotype across three SNPs upstream of *ctrF* (involved in capsule export), were more frequent in carriage ($p = 10^{-6.93}$, S3 Fig). The *ctrF* transcriptional start site has been mapped and the -10 box upstream of the gene predicted according to consensus sequences, although no -35 box was detected [35]. The sequence covered by the significant kmers revealed a match to four of the six nucleotides of the *E. coli* consensus sequence for the $\sigma^{70}$–10 Pribnow box and a match to three out of six nucleotides for the -35 sequence, separated by 17 bp, the optimal promoter spacing (S3 Fig). The SNPs captured by the significant kmers were within the promoter spacer at positions -14, -19 and -28, and the second most common IMD associated alleles were all Ts (S3 Fig). Mutating the *Plac* promoter spacer sequence from GC-rich to AT-rich has previously been shown to make it hyperactive [36] and the spacer sequence and the length of poly-T tracts has also been shown to influence DNA bending and promoter activity in *E. coli* [37,38]

Many bacterial species including *N. meningitidis* contain an extended -10 promoter, which is a '5-TRTG-3' sequence (where R is A or G) positioned one nucleotide upstream of the -10 box [35,39,40]. The extended -10 region has been shown to strengthen the promoter activity by enhancing the interaction between the promoter and RNA polymerase-sigma factor transcription initiation complex [41]. Mutagenesis analysis of *E. coli* promoters has revealed that substitutions in extended -10 regions differentially affect promoter activity [42]. Interestingly, the *ctrF* promoter contains a putative extended -10 sequence (S3 Fig), harbouring the SNP at position -14. Therefore, we hypothesise that the three SNPs in the *ctrF* spacer region impact promoter activity and expression of *ctrF*.

## Meningococcal disease associated island gene *tspB*

Within the MDA island, nine carriage-associated kmers tagged SNPs in region of *tspB*, deduced to encode the IgG binding domain of the TspB protein ($p = 10^{-8.51}$; S4 Fig) [30,31]. The nine kmers aligned perfectly to one of three copies of *tspB* in the isolate H44/76 (nmbh4476_0681), the copy shown to be most important for resistance to normal human serum (NHS) [31], in the aminoterminal domain of the highly conserved region which binds to IgG. The significant kmers altogether cover nine SNPs, of these two synonymous SNPs differentiate the H44/76 *tspB* copy important for resistance to NHS from the other two copies. Therefore, the kmers may be capturing the H44/76 nmbh4476_0681 allele of *tspB*.

## Other loci

Several other loci contained significant hits (Table 1), including a band of SNPs in perfect linkage disequilibrium in six genes ($p = 10^{-7.10}$; Fig 1G–1M). These were (i) non-synonymous SNPs in *gidA*, *mutL* and *lptF*; (ii) synonymous SNPs in *trkH* and *NMC_RS06715*; and (iii) an intergenic SNP between *NMC_RS11425*, pseudogenised in FAM18, and *vacB*. Significant kmers captured the same band of variants in LD, with the addition of kmers more significant in the intergenic region between *NMC_RS11425* and *vacB* ($p = 10^{-7.92}$). Additional variants captured only by kmers were identified in the genes *ctrG*, *NMC_02300*, *pilV*, *frpC* operon gene *NMC_02835*, pseudogenes *NMC_12405* and *mspA*, and *mafB-CT* ($p < 10^{-6.94}$).

## Polymorphisms in the *fba-fHbp* operon

We identified novel signals in a region of elevated significance within the *fba-fHbp* operon (Fig 2A). The *fba* gene (NEISS0350, annotated as *cbbA* in some meningococcal genomes, Table 1) encodes fructose-1,6-bisphosphate aldolase (Fba), which functions in carbon

**Table 1. Summary of significant k-mer associations.** The number of significant kmers, most significant -$\log_{10}$ p-values and $\beta$ point estimates for the most significant kmers for each gene. (p) denotes pseudogenes, IR = intergenic region. * N/A: not applicable as FAM18 is a serogroup C isolate and this locus is occupied in FANM18 by polysialyl-transferase gene *csc* (NMC0051).

| Locus/ gene | NEISS number | Annotation in FAM18 reference genome [25] | Encoded function | Ref | Number of significant k-mers | -$\log_{10}$ p-values | $\beta$ -point estimates |
|---|---|---|---|---|---|---|---|
| *tspB* | NEIS0025 | NMC0025 | Immune interactions | [30] | 9 | 8.51 | -0.49 |
| *ctrG* | NEIS0049 | NMC0049 | Capsule transport | [93] | 45 | 7.41 | 0.43 |
| *csb* | NEIS2161 | N/A* | Capsule synthesis | [32] | 21 | 8.58 | 0.45 |
| *ctrE-ctrF* IR | NEIS0066-NEIS0067 | NMC0066-NMC0067 | Intergenic region | [32] | 9 | 6.93 | -0.54 |
| *gidA* | NEIS0184 | NMC0184 | tRNA modification | [25] | 31 | 7.1 | -0.59 |
| *fba* (*cbbA*) | NEIS0350 | NMC0350 | Metabolism/ moonlighting protein | [43] | 2 | 7.16 | 0.59 |
| NMC_RS02300 | - | NMC_RS 02300 | Hypothetical protein | [25] | 31 | 7.1 | -0.59 |
| *pilV* | NEIS0487 | NMC0487 | Pilin protein | | 25 | 6.94 | 0.72 |
| *frpC* operon protein | NEIS0526 | NMC0526 | Hypothetical protein | [25] | 4 | 7.45 | -0.48 |
| *trkH* | NEIS0609 | NMC0609 | Potassium transporter | [61] | 62 | 7.1 | 0.59, -0.59 |
| NMC_RS11425 (p) -*vacB* IR | NEIS1102 | NMC1102 | Putative ribonuclease Intergenic region | [25] | 48 | 7.92 | 0.58 |
| NMC_RS12405 (p) | NEIS1156 | NMC1156 | Hypothetical protein | [25] | 22 | 10.93 | -0.59 |
| NMC_RS06715 | NEIS1285 | NMC1285 | Hypothetical protein | [25] | 31 | 7.1 | -0.59 |
| *mutL* | NEIS1378 | NMC1378 | Mismatch repair | [65] | 31 | 7.1 | -0.59 |
| *lptF* | NEIS1490 | NMC1490 | LPS transport | [57] | 39 | 7.1 | -0.59 0.59 |
| *mspA* (p) | NEIS1974 | NMC1974 | Auto transporter | [60] | 2 | 8.11 | -0.50 |
| *mafB-CT* | NEIS2090 | NMC2090 | Secreted toxin | [61] | 53 | 7.2 | 0.39 |

metabolism and cell adhesion [43,44], while *fHbp* encodes fHbp. The *fba-fHbp* signals were: (i) independent of the six other genome-wide significant SNPs; (ii) physically localised in a single region, unlike other polymorphisms; and, (iii) displayed a significant decay of signal around a prominent peak, characteristic of an authentic association (Figs 1D–1M and S5). The significant IMD-associated SNP ($P = 10^{-6.51}$) occurred at high frequency in the sample (58.7% invasive cases, 17.2% carriers), and explained 10% of sample heritability. Therefore, while several signals were detected across the genome, the association at *fba-fHbp* was of particular interest.

The significant SNP in the *fba-fHbp* locus occurred at nucleotide 900 of *fba* ($fba_{S900}$, $P = 10^{-6.51}$), near the 3′ end of *fba*, as did two kmers spanning this SNP commencing at nucleotides 898 and 899 of *fba* ($fba_{K898}$ and $fba_{K899}$; $p = 10^{-7.16}$). There was a genome-wide significant enrichment in the rest of the *fba-fHbp* operon (adjusted harmonic mean $p = 10^{-1.72}$). The two kmers spanned protein-coding nucleotides 898–929, tagging the $fba_{S900}$ SNP and two others: $fba_{S912}$ ($p = 10^{-2.25}$) and $fba_{S913}$ ($p = 10^{-2.25}$) (Figs 2A and S6). The peak signal of association therefore coincided with $fba_{S900}$, which was in tight linkage disequilibrium with a neighbouring SNP $fba_{S897}$ ($P = 10^{-5.77}$, $r^2 = 0.98$). $fba_{S900}$ and $fba_{S897}$ both cause synonymous substitutions, located 323 and 326 bp upstream of the *fHbp* start codon, respectively.

A replication study was undertaken to test the association of $fba_{S900}$ with IMD using genomes of an extended set of 1,295 cc41/44 meningococci, comprising 1046 IMD and 249 carriage isolates (available at https://PubMLST.org/neisseria [45]) (S7 Fig). The cc41/44 meningococci are a leading cause of IMD world-wide [46], and are polymorphic at $fba_{S900}$. Analysing a single clonal complex mitigates confounding due to heterogeneous sampling across diverse lineages [47,48]. After Bonferroni correction for two candidate kmers ($p<10^{-1.60}$), the IMD-associated signal from kmers $fba_{K898}$ and $fba_{K899}$ was replicated in the cc41/44 isolates ($p = 10^{-2.37}$), with the

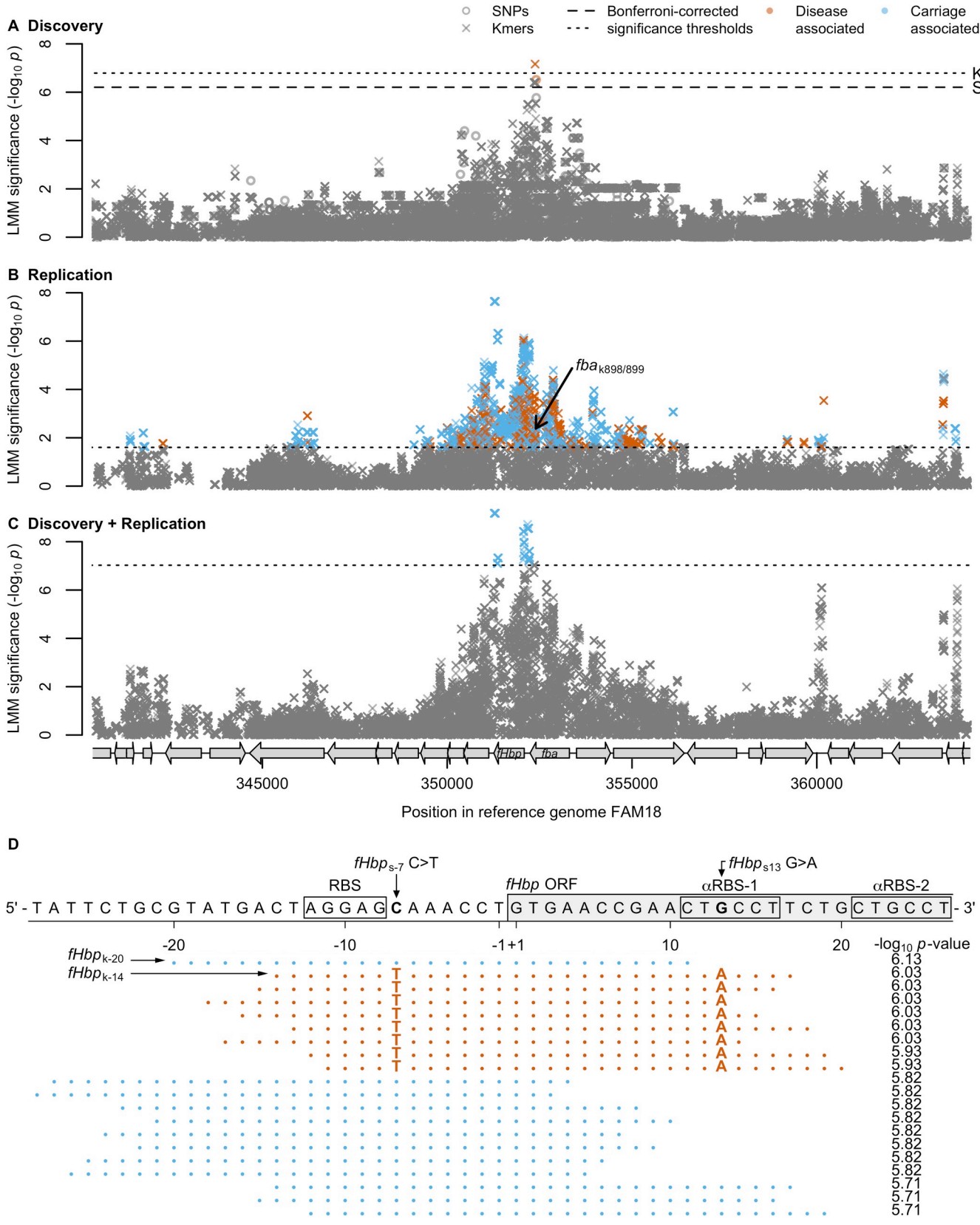

**Fig 2. (A-C)** SNPs and kmers in the *fba-fHbp* operon plus the surrounding 10kb. Open circles represent SNPs and crosses represent left-most kmer mapping positions. Significant SNPs and kmers are coloured by the LMM estimated direction of effect. Bonferroni-corrected significance thresholds are shown by black dashed lines (SNPs) or black dotted lines (kmers). Grey arrows represent coding sequences in the reference genome FAM18. (A) The discovery 261 isolates sampled from the Czech Republic in 1993. (B) Kmers above 1% minor allele frequency (MAF) in 1,295 ST-41/44 complex replication study genomes. The black arrow points to the position and significance of the two kmers in the gene *fba* that were significant in the discovery sample collection. (C) Kmers above 1% (MAF) in the discovery 261 Czech Republic sample collection plus the 1,295 ST-41/44 complex replication study genomes. (D) The 20 most significant kmers in the replication study surrounding the *fHbp* start codon. The FAM18 reference genome is shown for positions 352112–352059. The start of the open reading frame (ORF), the ribosomal binding site (RBS) and two putative anti-RBS sites are annotated. Kmer sequences are depicted by dots where they are the same as the reference, and by their base where they differ. Blue kmers are estimated to be associated with carriage, and dark orange kmers with invasive disease. The kmer $-\log_{10}$ *p*-values are annotated. Of the top 20 kmers in this region, the carriage-associated kmers were identical to FAM18, and the disease-associated kmers contained two annotated SNPs, a T at fHbp position -7, 1 bp away from the RBS, and an A at position 13 within the putative anti-RBS-1 sequence.

direction of the effect replicated (β = 0.16). Moreover, the general enrichment in significance in *fba-fHbp* was replicated (adjusted harmonic mean $p = 10^{-1.96}$).

We explored possible effects of the synonymous SNPs in *fba* on the expression of *fHbp*, which can be translated from a bicistronic *fba-fHbp* mRNA or from a *fHbp*-specific promoter [49]. Expression of *fHbp* can be regulated by FNR binding to sequences 80 bp upstream of the start codon [49]. We noticed that the synonymous IMD-associated substitutions at $fba_{S897}$ and $fba_{S900}$ form a motif resembling an FNR box 314 bp upstream of *fHbp* (Figs 3A and S6). Electrophoresis mobility shift assays (EMSA) demonstrated binding of a constitutively active version of FNR to the known FNR site but not to sequences within *fba*, irrespective of the SNP sequence (Figs 3B and S8). Furthermore, there was no detectable difference in fHbp expression by four isogenic constructs of a cc41/44 isolate, covering all combinations of the SNPs $fba_{S897}$C/T and $fba_{S900}$T/C (Fig 3C and 3D).

We considered whether other variants in the *fba-fHbp* region could be driving the signals of association, as a total of 1,346 kmers in *fba-fHbp* were more significant in the replication study than the candidate kmers, $fba_{K898}$ and $fba_{K899}$ (Fig 2B–2D). The most significant kmers above 1% minor allele frequency in the replication study were those starting at *fHbp* nt -20 and -14 (henceforth $fHbp_{K-20}$ and $fHbp_{K-14}$, respectively $p < 10^{-5.93}$), at nt 686 ($fHbp_{K686}$, $p < 10^{-6.04}$), and nt 752 ($fHbp_{K752}$, $p < 10^{-7.64}$), relative to the first base of the start codon. The SNPs tagged by these kmers were: (i) $fHbp_{S-7}$C/T in the 5'-untranslated region (5'-UTR) of *fHbp* adjacent to the ribosome binding site (RBS, $p = 10^{-6.13}$); (ii) $fHbp_{S13}$G/A encoding an Ala⁵Thr substitution in *fHbp* near a previously identified anti-RBS (α-RBS) site ($p = 10^{-6.03}$) [50]; (iii) $fHbp_{S781}$A/G which leads to an Arg²⁶¹Gly substitution in fHbp adjacent to the CFH binding site ($p = 10^{-7.64}$); and (iv) $fHbp_{S700}$G/A causing a Gly²³⁴Ser substitution distant from the site of CFH ($p = 10^{-6.32}$) (Figs 3E and S9–S11).

The IMD-associated SNP $fHbp_{S781}$G removes a charged side chain (on Arg²⁶¹), which could affect interactions with CFH (Fig 3E). We generated proteins with Arg²⁶¹ or Gly²⁶¹ in v2.24 fHbp, the allele most significantly associated with IMD ($p = 10^{-3.23}$, S12 Fig), and assessed fHbp:CFH binding by ELISA; however, we found no evidence that $fHbp_{S781}$ altered binding to CFH (Fig 3F).

To explore an alternative mechanism, we examined the effect of the SNPs around the RBS and α-RBS sequence using SHAPE chemistry to probe the RNA secondary structure using a 183 nucleotide RNA encompassing the RBS at position -8 to -12, relative to the translation start site, and $fHbp_{S-7}$T/C and $fHbp_{S13}$A/G. The secondary structure model based on SHAPE reactivity data of $fHbp_{S-7}$C/$_{S13}$G at 37˚C (Fig 4A) was consistent with the RBS being base-paired and masked through the formation of a relatively long imperfect helix of 11 base pairs that included both anti-RBS sequences 1 (αRBS-1) and 2 (αRBS-2) [50]; the polymorphic sites in the carriage associated $fHbp_{S-7}$C/$_{S13}$G structure formed a G:C base pair at the top of the helix. However, the local RNA structure of the IMD-associated $fHbp_{S-7}$T/$_{S13}$A showed

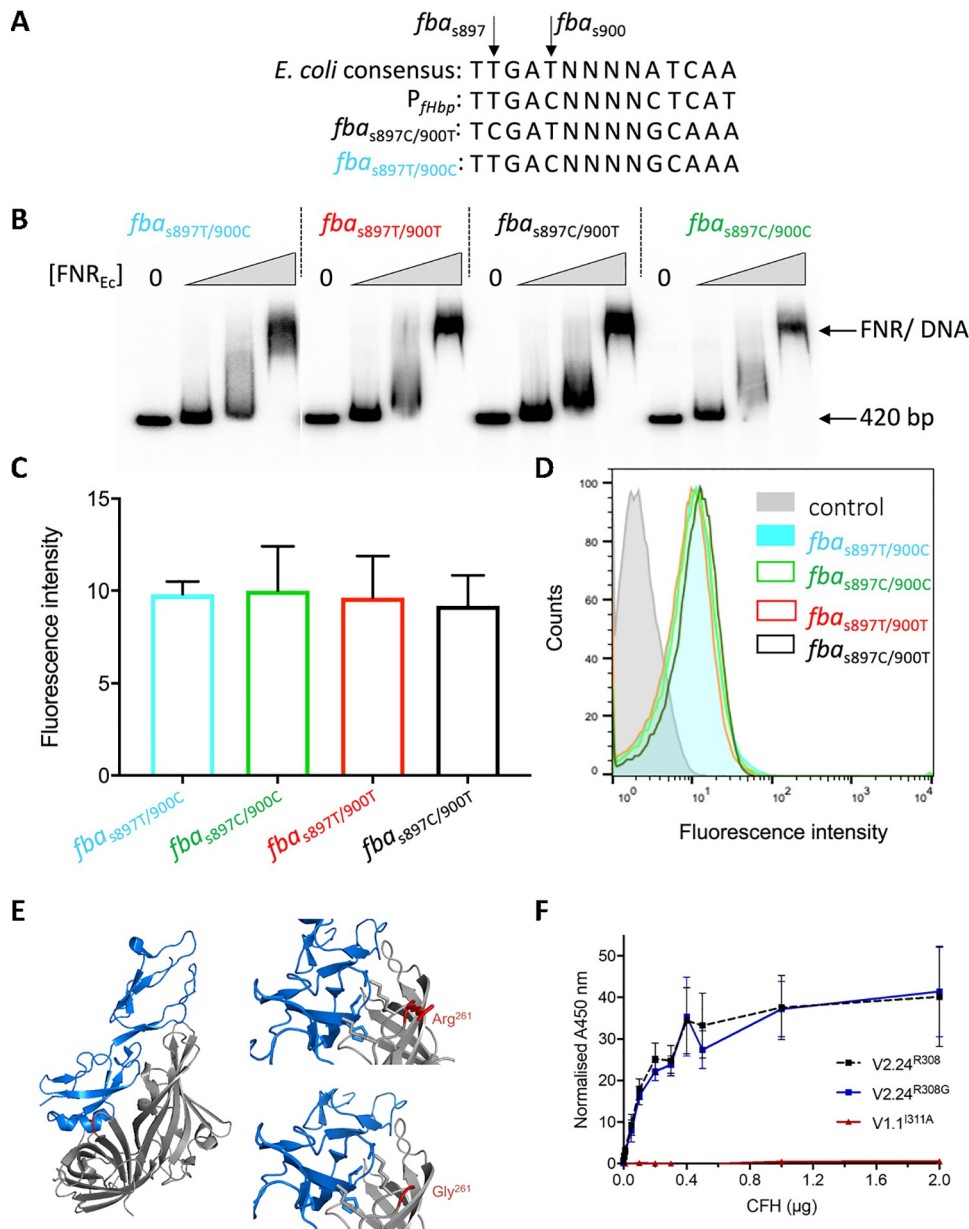

**Fig 3. SNPs in *fba* do not alter FNR binding or fHbp surface expression.** **(A)** Sequences of SNPs in *fba* aligned with the consensus *E. coli* FNR binding sequence and the known FNR site upstream of *fHbp*. **(B)** EMSA of 420 bp upstream of *fHbp* (including the known FNR binding site) with increasing concentrations of FNR (0, 0.75, 1.5, 3 μM). **(C)** fHbp was detected on the surface of bacteria with different SNPs (indicated) by flow cytometry using α-fHbp pAbs. Geometric mean fluorescence was used to compare fHbp levels across the samples. Error bars show SEM, significance analysed by two-way ANOVA showed no statistical difference between the strains. **(D)** Representative flow cytometry histograms; strains indicated; control (grey filled area), no primary pAb. **(E)** Side chains of Arg[261] and Gly[261] (red) of fHbp (grey) shown with CCPs 6 and 7 of CFH (blue) and threaded onto fHbp (v3.28, PDB:4AYI); figures generated in PyMOL. **(F)** Binding of fHbps to CFH by ELISA; a non-functional fHbp (v1.1 Ala[311]) was included as a control; error bars, SD, n = 3.

significant differences (Fig 4B and 4C), with the 6 bp structure around the RBS being much more open and accessible (S13 and S14 Figs for SHAPE analysis and predicted RNA structures at 30˚C and 42˚C). These data demonstrated that the RNA structure around the RBS was

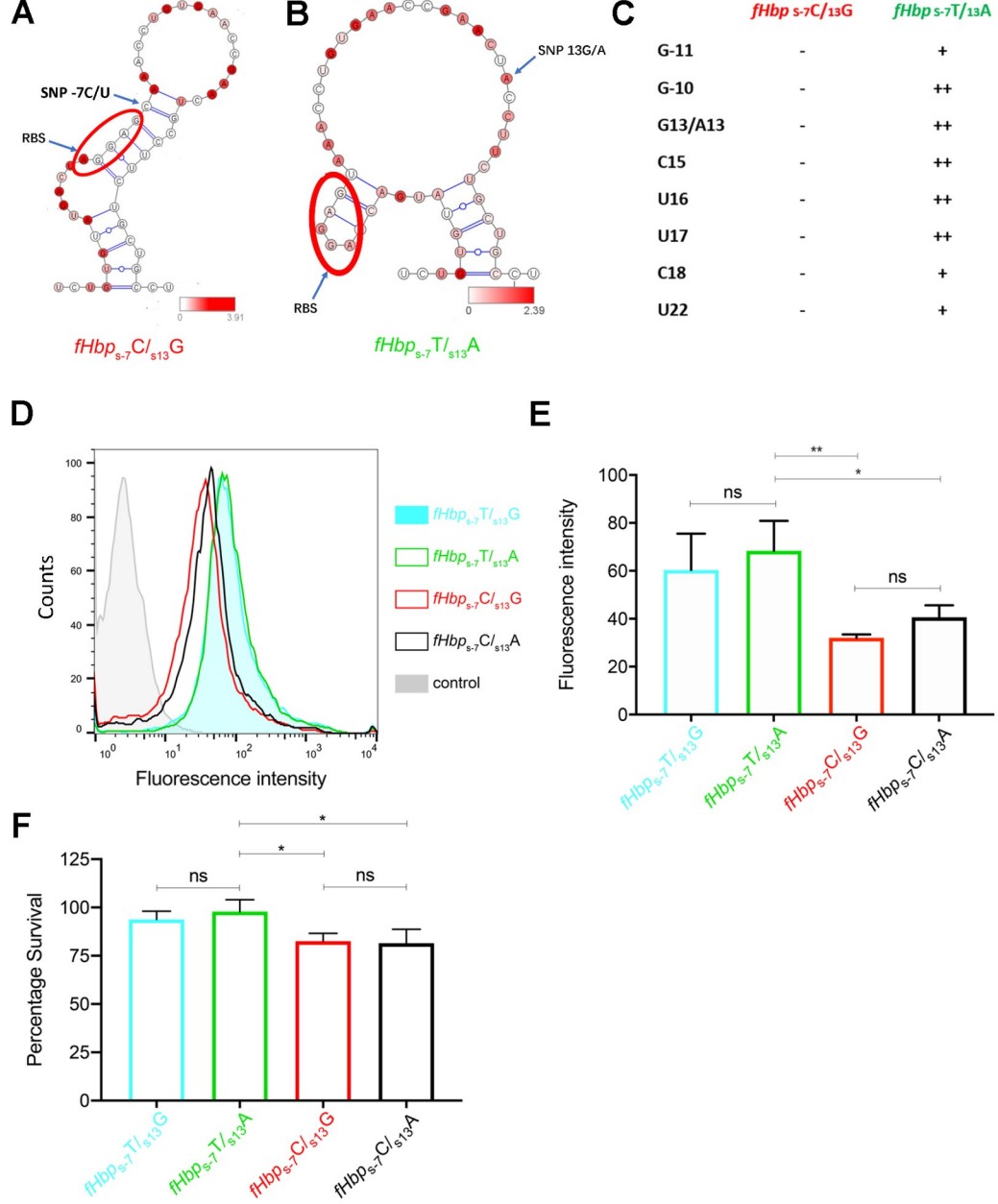

**Fig 4. Secondary structure of the RNA structure at 37˚C around the RBS calculated using RNAstructure and SHAPE reactivity data. (A)** $fHbp_{s-7}C/fHbp_{s13}G$ and **(B)** $fHbp_{s-7}T/fHbp_{s13}A$; SHAPE reactivity data are mapped on the RNA structure and colour coded by intensity as shown on the bars; the RBS is circled in red **(C)** Nucleotides with reactivity are listed in the table as strong (++), medium (+) and weak (-). Representative flow cytometry histograms and geometric mean fluorescence **(D, E)** of surface fHbp on *N. meningitidis* with SNPs $fHbp_{s-7}T/fHbp_{s13}G$ (blue filled area), $fHbp_{s-7}T/fHbp_{s13}A$ (solid green line), $fHbp_{s-7}C/fHbp_{s13}G$ (solid red line) or $fHbp_{s-7}C/fHbp_{s13}A$ (solid black line). Bacteria were grown at 37˚C, and fHbp detected with anti-fHbp pAb; control (grey filled), bacteria with no primary antibody. **(F)** Serum sensitivity assays of *N. meningitidis* strains with SNPs as indicated. Error bars show SD (n = 3), * $p<0.05$, ** $p<0.01$ (n = 3, two-way ANOVA).

modulated by sequence variation, suggesting that the polymorphisms modulate initiation of protein synthesis.

To examine the impact of the polymorphisms on fHbp expression, we constructed a series of isogenic cc41/44 mutants with combinations of the SNPs, $fHbp_{S-7}T/C$ and $fHbp_{S13}A/G$, and

examined surface expression of fHbp. Notably, the $fHbp_{S-7}$T IMD risk allele conferred significantly higher fHbp expression, measured by flow cytometry, than $fHbp_{S-7}$C, irrespective of $fHbp_{S13}$A/G ($p<0.05$, Fig 4D and 4E). Further, when bacteria were incubated in normal human sera (NHS), strains with $fHbp_{S-7}$T displayed increased survival compared with $fHbp_{S-7}$C, but not in heat-inactivated human serum lacking functional complement, irrespective of the $fHbp_{S13}$A/G allele ($p<0.05$, One way Anova, Figs 4F and S15). Taken together, these results were consistent with the IMD-associated alleles at the 5′ end of *fHbp* conferring enhanced resistance of bacteria against complement-mediated killing, a major component of immunity against *N. meningitidis*.

## Discussion

The GWAS approach employed here identified meningococcal genetic variants associated with IMD by comparing disease and carriage isolates, while controlling for population structure. The study exclusively used publicly available genome sequences and metadata stored in the PubMLST *Neisseria* database (https://pubmlst.org/neisseria/), using well-described datasets from the Czech Republic and of cc41/44 isolates for replication. GWAS studies of virulence are particularly suitable in recombinogenic organisms such as *N. meningitidis*, as recombination assists fine-mapping by breaking down clonal background [20,22,51–53]. Previous GWASs have not identified variants influencing IMD severity, including in *fHbp*, and adaptation has also not been identified between paired isolates sampled from blood and cerebrospinal fluid [54,55]. We confirmed that the genetic architecture of meningococcal virulence is polygenic, identifying significant associations with IMD *versus*. carriage across the genome. Of note, we found that among others, polymorphisms at the capsule gene *csb* and MDA island gene *tspB* contribute to the invasiveness of strains in our discovery studies, adding to the growing understanding on virulence factors influencing the risk of IMD [15,28,29,56].

Most of the significant associations found (Fig 1 and Table 1) were with genes or genome regions that either: (i) encoded the production of meningococcal surface and extracellular structures that interact with host cells and/or immune molecules; or (ii) were plausibly associated with expression of genes encoding these structures. The capsular polysaccharide region (*cps*), encoding the best understood meningococcal virulence factor [32], was tagged by associations with the *csb* and *ctrG*, genes and the *ctrE-ctrF* intergenic region, all of which were consistent with expression of the capsule, well known to be important in invasion. The *lptF* gene encodes LptF, an inner membrane protein involved in the transport of lipopolysaccharide (LOS) onto the meningococcal outer membrane [57]. LOS has well-established roles in host-pathogen interactions including immune evasion [58]. The *pilV* gene was also identified: this gene encodes PilV, part of the type IV pilus (Tpf), an adhesin that binds to host endothelial and epithelial cells [59].

MspA, encoded by the *mspA* gene, is an autotransporter known to elicit immune responses in humans, which is also known to act as an adhesin to human cells and is variably present among hyperinvasive meningococci, being found in some but not all cc11 isolates [60]. The phage-associated T and B cell-stimulating protein (TspB), encoded by *tspB*, is an immunoglobulin binding protein, conferring serum resistance on meningococci as well as being implicated in the formation of biofilms [30,31]. The *mafB* gene occurs in *maf* genomic islands (MGI) that encode polymorphic toxins, which are essentially absent in non-pathogenic *Neisseria* [61]: these proteins probably play a role in adhesion to host cells [62]. FrpC, encoded by the iron regulated *frpC* gene, is a member of the RTX toxin family, which is secreted by meningococci in the early stages of infection, inducing high levels of serum antibody [63]. Other genes identified included *trkH*, encoding a potassium transporter, located within the *trk* operon flanking the 3' end of the MGI-2 island [61]. Potassium homeostasis is essential for bacterial survival

and has been implicated in the virulence of several bacterial pathogens [64]. The *vacB* gene, encoding RNAse R, and *gidA* gene are involved in RNA metabolism and in cell division. Mutations in the *mutL* gene, encoding the MutL mismatch repair protein, are associated with elevated rates of mutation characteristic of epidemic meningococci, probably because of increased slip-strand mispairing leading to variation in expression of contingency genes [65].

Of the signals associated with the invasive phenotype, we experimentally tested the strongest, which emanated from the *fba-fHbp* region. This was particularly noteworthy as GWAS of human genetic susceptibility identified SNPs in and around CFH/CFHR3 to be associated with IMD [11,66]. In addition to its proximity to the *fHbp* gene, the *fba* gene (PubMLST NEISS0350, annotated as *cbbA* in several meningococcal genomes), encodes fructose-1,6-bisphosphate aldolase (Fba), which is located on the meningococcal cell surface, mediating adherence to human cells [43]. Fba also has a possible 'moonlighting' function, binding to human glu-plasminogen [44]. As the polymorphisms identified in *fba* (*cbbA*) were synonymous, there was no obvious role for these polymorphisms in such interactions; however, *fba* (*cbbA*) is adjacent to the *fHbp* gene and both genes can be expressed from the same, or individual transcripts [49], the expression differing among different clonal complexes [67]. The regulation of these two transcripts is complex, with the *fba* promotor positively regulated by the presence of Oxygen and absence of iron while the converse is true for *fHbp* promotor [49]. The *fba-fHbp* intergenic region is variable and contains a Rho-independent binding site an FNR binding site [49] and an RNA thermosensor [68]. We showed that, while there was no evidence for the *fba* mutations influencing fHbp expression, the polymorphisms we identified in the *fba-fHbp* intergenic region modulated fHbp expression, which in turn affected meningococcal survival in human serum. The mechanism was likely due to changes in RNA structure caused by the polymorphisms identified.

Our findings were complementary with, but not identical to, a previous study by Spinsanti *et al.* of the *fba*(*cbbA*)-*fHbp* intergenic region (fIR) [19]. They constructed isogenic strains of eleven upstream alleles, plus isogenic strains of particular variants to demonstrate that fIR alleles were associated with particular levels of fHbp expression, with increased expression associated with serum resistance and the propensity to cause meningococcal disease (S1 Table) [19]. Of the alleles tested by Spinsanti *et al.*, allele fIR16 was identical to our isolate 0011 $fHbp_{s-7}$T/$_{s13}$G; however, fIR16 was part of a the bicistronic operon with *fba*, resulting in higher overall expression of fHbp. Other than at the polymorphism we term $fHbp_{s-7}$ the fIR3 and fIR13 sequences were identical, although both were categorised as low expression by Spinsanti *et al.*. Compared to our experimental background strain, these alleles had a different -10 box, defined in the Spinsanti *et al.* study as causing higher fHbp expression, plus polymorphisms at three further sites (S5 Table), therefore the difference in expression that we identified at $fHbp_{s-7}$ is perhaps only detectable with lower baseline levels of fHbp expression, dependent on the -10 box and fHbp variant. The highest expressor (fIR1) was the only allele with both of our tested upstream disease-associated variants, $fHbp_{s-7}$T and $fHbp_{s13}$A. This also had the same -10 box as our background strain and only one other polymorphism, although the allele was also found to be part of a bicistronic operon with *fba*.

Differences between the results of the two studies were likely due to differences in the meningococcal isolates and mutants used. Whereas our study tested mutants of isolate 0011/ 93, a serogroup B cc41/44 meningococcus containing fHbp variant 3, the Spinsanti *et al.* study used MC58 a serogroup B cc32 isolate as their background strain. Further, they tested alleles controlling fHbp variant 1 which is expressed at higher levels compared to fHbp variant 3 used in our study [19].

In conclusion, both studies agree that upstream variants affect fHbp expression and survival in human serum. In common with Spinsanti *et al.*, who used over 5,800 PubMLST isolates

sampled from the UK to show that higher predicted fHbp expression was associated with a higher proportion of invasive disease isolates, we also identified fHbp as associated with IMD using PubMLST data, with a 63% overlap in our replication isolates. Our initial PubMLST discovery data were, however, entirely independent, as they were sampled from the Czech Republic (see Materials and Methods for overlap of samples in our and other studies). Furthermore, in our discovery and validation studies we controlled for population structure, which was not done in the Spinsanti *et al.* study [19]. Direct comparison of the experimental results is complicated, as the variants tested and the strain backgrounds differed between the studies: we chose test isolate genomes belonging to the cc from which the polymorphisms were identified (S5 Table).

Although many cases of IMD can be prevented by conjugate polysaccharide vaccines, the absence of serogroup B polysaccharide vaccines remains a challenge. While several protein-based vaccines have been developed, none of these are comprehensive and their development has been hampered by the lack of an understanding of the nature of the invasive phenotype, exacerbated by a lack of suitable animal models and correlates of protection. Our GWAS approach identified a suite of genes that plausibly contribute to the invasive phenotype many of which encoded proteins implicated in immunity to IMD. Among other IMD-associated variants, we identified the involvement of individual upstream regulatory variants in *fHbp* in disease status by taking a hypothesis-free, genome-wide approach that controlled for population structure. This work was independent of and complimentary to the candidate gene approach [19] and confirmed and enhanced the importance of individual variants upstream of *fHbp* as leading genome-wide variants using independent discovery data. Whilst supporting the contention that the meningococcal hyperinvasive phenotype is polygenic, our results further strengthen the concept that host susceptibility, and the propensity of meningococci to cause invasive disease, is dictated by the interaction of bacterial and human gene pools across a single molecular interface, that between fHbp and CFH. Although this study highlights the complexity of host-meningococcal interactions at the molecular level, as they become better defined it will become increasingly possible to make genome-based predictions of the likely clinical and epidemiological properties of meningococcal variants, enhancing our capacity to combat IMD with vaccination and other interventions.

## Materials and methods

### Sampling frames

The discovery sample collection comprised 261 *Neisseria meningitidis* isolates from the Czech Republic [20–22]. Carriage samples were collected from young adults with no association with patients with IMD over four months, while disease isolates were from cases of IMD submitted to the Czech National Reference Laboratory for Meningococcal Infections in 1993 [20,22]. Illumina sequencing reads were downloaded from the European Nucleotide Archive (ENA, http://www.ebi.ac.uk/ena), and Velvet *de novo* assemblies from PubMLST (https://pubmlst.org/neisseria/) [45]. PubMLST IDs for isolate records, along with ENA accession numbers for sequence data, and isolate phenotypes are provided in S6 Table.

The replication sample collection comprised 1,295 genomes from cc41/44 isolates downloaded from PubMLST. We downloaded all genomes 10th August 2018 with a non-empty disease or carriage status, with *de novo* assemblies ≥2Mb in length, excluding the 261 genomes from the discovery sample collection and also excluding genomes that met any of the following criteria: (i) annotated as non-*Neisseria meningitidis*; (ii) annotated with the disease phenotype "other"; (iii) non-ST-41/44 complex assignment (described below); (iv) genomes with more

than 700 contigs; (v) genomes with only one or two contigs; and, (vi) genomes with a total assembly length greater than 2.5Mb. PubMLST IDs and phenotypes can be found in S7 Table.

A total of 35 of 261 of the discovery samples were previously used in the microarray-based discovery of the Meningococcal Disease Associated (MDA) island [28] with MDA presence validated by PCR. A total of 252 of 261 discovery samples were used in the PCR-based validation of MDA presence [28], with 64/261 discovery and 2/1295 validation samples used in a serogroup C WGS-based GWAS [56]. Finally, 818/1295 validation samples were used in the *fHbp* candidate gene analysis [19].

## SNP calling and kmer counting

For the discovery sample collection, sequence reads were mapped against the cc11 reference genome from isolate FAM18 (GenBank number: NC_008767.1) using Stampy [69]. Bases were called using previously described quality filters [70–72]. We identified 150,502 biallelic SNPs, 6,063 tri-allelic SNPs, and 239 tetra-allelic SNPs.

To capture non-SNP-based variation, and SNPs not in the reference genome, we pursued a kmer-based approach, where all unique 31 bp haplotypes were counted from Velvet assemblies using dsk [73] in both sample collections. For both sample collections, a unique set of variably present kmers across each data set was created, with the presence or absence of each unique kmer determined per genome. Algorithms, coded in C++, can be downloaded from https://github.com/jessiewu/bacterialGWAS/blob/master/kmerGWAS/gwas_kmer_pattern [27]. We identified 7,806,583 variably present kmers in the discovery sample collection and 11,114,868 in the replication study collection.

## Phylogenetic inference

A maximum likelihood phylogeny was estimated for the discovery study collection for visualisation of phylogenetic relationships in the sampling frame and for SNP imputation purposes using RaxML [74] with a general time reversible (GTR) model and no rate heterogeneity, using alignments from the mapped data based on biallelic sites, with non-biallelic sites being set to the reference allele.

Non-cc41/44 assignment in the replication study collection was determined using the kmer counts. A UPGMA tree was built using a kmer presence/absence distance matrix and all descendants of the most recent common ancestor of the genomes annotated in PubMLST as cc41/44 were kept in order to identify unlabelled members of the complex.

## SNP imputation

For the SNP discovery analysis, missing sites due to sequencing ambiguity or strict SNP-calling thresholds were imputed using ancestral state reconstruction [75] implemented in ClonalFrameML [76]. This approach was previously shown to achieve high accuracy [27].

## Correcting for multiple testing

Multiple testing was accounted for by applying Bonferroni correction to control the strong-sense family-wise error rate (ssFWER) [77]. The unit of correction for all studies of individual loci in the discovery GWAS was taken to be the number of unique "phylopatterns" *i.e.* the number of unique partitions of individuals according to allele membership. The locus effect of an individual variant was considered to be significant if its *p*-value was smaller than $\alpha/n_p$, where we took $\alpha = 0.05$ to be the genome-wide false positive rate (*i.e.* family-wise error rate, FWER) and $n_p$ to be the number of unique phylopatterns. In the discovery SNP-based analysis,

$n_p$ was taken to be the number of unique SNP phylopatterns (80,099) and in the kmer-based analyses, $n_p$ was taken to be the number of unique kmer presence/absence phylopatterns (307,830).

In the replication GWAS, since we tested whether the two genome-wide significant, disease-associated kmers in the *fba-fHbp* operon replicated, we applied a Bonferroni correction to obtain a significance threshold of $0.05/2 = 10^{-1.60}$.

When testing the discovery GWAS for lineage effects by the Wald test for principal component-phenotype associations, a Bonferroni correction was applied for the number of non-redundant principal components, which equalled the sample size (261) since no two genomes were identical.

## Testing for locus effects

We performed association testing using the R package *bugwas* (https://github.com/sgearle/bugwas) [27] which uses linear mixed model (LMM) analyses implemented in the software GEMMA [24] to control for population structure. We modified GEMMA version 0.93 to enable significance testing of non-biallelic variants [27]. GEMMA was run using no minor allele frequency cut-off to include all variants.

For the discovery GWAS, we computed the relatedness matrix from biallelic SNPs only using the option "-gk 1" in GEMMA to calculate the centred relatedness matrix. For the replication study, we computed the relatedness matrix from the kmer presence/absence matrix using Java code which also calculates the centred relatedness matrix.

## Identifying lineage effects

We tested for associations between bacterial lineages and the phenotype in the discovery sample collection using the R package *bugwas* (https://github.com/sgearle/bugwas) [27]. Principal components were computed based on biallelic SNPs and interpreted in terms of bacterial lineages. To test the null hypothesis of no background effect of each principal component we employed a Wald test, where we compared the test statistic against a $\chi^2$ distribution with one degree of freedom to obtain a *p*-value.

## Estimating sample heritability

Heritability of the sample, the proportion of the phenotypic variation that can be explained by the bacterial genotype, was estimated using the LMM null model in GEMMA from post-imputation biallelic SNPs [24] Estimating heritability in case control studies is dependent on the prevalence of cases and the sampling scheme [78]. The proportion of sample heritability explained by the kmers $fba_{K898}$ and $fba_{K899}$ in the discovery set was estimated by including the phylopattern of the two kmers as a covariate in the LMM null model in GEMMA and calculating the difference in heritability compared to including no covariates.

## Testing for independent SNP associations

To determine whether pairs of significant SNP associations in the discovery sample collection represented independent signals, the two unique significant SNP patterns were tested using LMM including both SNPs as fixed effects, thereby assuming additivity between the two loci.

## Variant annotation

SNPs were annotated in R using scripts at http://github.com/jessiewu/bacterialGWAS. The reference FASTA and GenBank files were used in order to determine SNP type (synonymous,

non-synonymous, nonsense, read-through, and intergenic), codon, codon position, reference and non-reference amino acids, gene name and gene product, on the assumption of a single change to the reference sequence.

To annotate kmer sequences, we mapped kmers to the reference FAM18 genome using Bowtie2 [79] and the options "-r -D 24 -R 3 -N 0 -L 18 -i S,1,0.30" to identify a single best mapping position for each kmer. For kmers which did not map to the reference genome, BLAST [80] was used to identify the kmer position within the FAM18 genome sequence. BLAST results of any sequence length were taken, and the number of mismatches along the whole length of the kmer was recalculated assuming the whole kmer aligned. Kmers with five or fewer mismatches to the reference were shown as aligned to the reference, all other kmers were shown as unaligned to the reference.

To understand the variation captured by the significant kmers in the gene *csb*, BLAST [80] was used to extract all copies of the MC58 (Genbank accession number NC_003112.2) allele of *csb*, the allele that the significant *csb* kmers mapped to.

As the reference FAM18 genome contains multiple copies of the gene *tspB*, to understand the variation captured by the significant kmers in *tspB*, BLAST [80] was used to identify all kmer alignments with just the FAM18 *tspB* gene NMC_RS00140.

## Harmonic mean *p*-value

The harmonic mean *p*-value (HMP) method performs a combined test of the null hypothesis that no *p*-value is significant [81]. The HMP method controls for the ssFWER like the Bonferroni correction. We applied the HMP procedure to the *fba-fHbp* region in the discovery and replication studies, including all unique kmer phylopatterns that mapped to either of the two genes plus their upstream intergenic regions. We calculated the asymptotically exact *p*-values using the p.hmp function from the R package 'harmonicmeanp', giving equal weight to all kmer phylopatterns, and the total number of tests performed genome-wide was set to be the number of kmer phylopatterns ($n_p$) in order to control the genome-wide ssFWER despite analysing just the *fba-fHbp* region. We adjusted the *p*-value by dividing it by the sum of the weights of the kmer phylopatterns included in the *fba-fHbp* region so that it could be directly compared to alpha, the intended ssFWER, which we set to be 0.05.

## Software

Software applied within these analyses can be found at http://github.com/jessiewu/bacterialGWAS and http://github.com/sgearle/bugwas.

## Strain construction

The primers and strains used to test the effects of SNPs are listed in S2–S4 Tables. The $fba_{S897}/_{S900}$ SNPs were constructed by inserting a Kanamycin resistance cassette downstream of *fHbp*. First, the upstream fragment (starting 843 bp upstream of the *fHbp* start codon including the C terminus of *fba*, terminating 12 bp downstream of the *fHbp* stop codon) and downstream fragment (751 bp downstream of the *fHbp* stop codon) were amplified with primers ERS001/ERS004 and ERS007/ERS008 respectively from 0011/93 *N. meningitidis* gDNA. The kanamycin resistance cassette was amplified from pGEMTEasy-Kan using ERS005/ERS006 and the three fragments were cloned into pUC19 using NEB Builder HiFi DNA assembly kit (New England Biolabs). A second set of overlap primers were used to introduce SNPs into a second upstream fragment using primer combinations: ERS001/ERS002 and ERS003/ERS004, ERS001/ERS009 and ERS010/ERS004, and ERS001/ERS011 and ERS012/ERS004. The constructs were purified and transformed into 0011/93 *N. meningitidis*. For each

strain, three independent single colonies were pooled and gDNA from the pooled stocks was checked by PCR and sequencing.

The *fHbp*$_{S-7}$/$_{S13}$ SNPs were constructed by inserting an erythromycin resistance cassette downstream of *fHbp*. First, a fragment corresponding to 496 bp upstream of the *fHbp* start codon and the *fHbp* ORF, and a fragment corresponding to 707 bp downstream of the *fHbp* stop codon) were amplified with primers ML428/ML429 and ML434/ML433 respectively from 0011/93 *N. meningitidis* gDNA. The erythromycin resistance cassette was amplified from pNMC2 [82] using ML430/ML435 and the three fragments were cloned into pUC19 using NEB Builder HiFi DNA assembly kit (New England Biolabs). The resulting vector was used as a template to generate *fHbp* with different SNPs by site directed mutagenesis using primer combinations: ML436/405 and ML437/406, ML438/ML405 and ML439/ML406, and ML440/ML405 and ML441/406. The constructs were purified and used to transform 0011/93 *N. meningitidis*. For each strain, three independent single transformants were pooled and gDNA from the pooled stocks was checked by PCR and sequencing.

## Generation of plasmids and protein purification

V2.24 *fHbp* was amplified from *N. meningitidis* OX99.32412 and SNPs introduced by PCR, then ligated into pET21b using Quick-Stick Ligase (Bioline). Versions of *fHbp* were ligated into pET28a-His-MBP-TEV (in frame with sequence encoding a histidine tag and the *Escherichia coli* maltose-binding protein (MBP) with a C-terminal TEV cleavage site) linearised with *Xho*I, and constructs confirmed by sequencing.

v2.24 fHbps were expressed in *E. coli* B834 during growth at 22˚C for 24 hrs with 1 mM IPTG (final concentration). Bacteria were harvested and resuspended in Buffer A (50 mM Na-phosphate pH 8.0, 300 mM NaCl, 30 mM imidazole) and the fHbp purified by Nickel affinity chromatography (Chelating Sepharose Fast Flow; GE Healthcare). Columns were washed with Buffer A, then with 80:20 Buffer A:Buffer B (50 mM Na-phosphate pH 8.0, 300 mM NaCl, 300 mM Imidazole), and proteins eluted in 40:60 Buffer A:Buffer B. Proteins were dialysed overnight at 4˚C into PBS, 1mM DTT pH 8.0 with TEV protease prior to Nickel affinity chromatography to remove the HIS-GST-TEV. fHbp was eluted from Sepharose columns with Buffer B after washing with buffer C (50 mM Na-phosphate pH 6.0, 500 mM NaCl, 30 mM Imidazole), and dialysed overnight at 4˚C into Tris pH 8.0. fHbp v1.1$^{I311A}$ expression and purification was performed as described previously [83].

## Electrophoresis mobility shift assays

Gel retardation assays were carried out as previously using purified FNR$^{D154A}$, which forms functional FNR dimers under aerobic conditions [84]. Sequences upstream of *fHbp* were amplified with primers ERS012/013, and the full length (420 bp) or *Hae*III-digested (294 and 126 bp) fragments end-labelled with [γ-$^{32}$P]-ATP with T4 polynucleotide kinase (New England BioLabs). Approximately 0.5 ng of each labelled fragment was incubated with varying amounts of FNR$^{D154A}$ in 10 mM potassium phosphate (pH 7.5), 100 mM potassium glutamate, 1 mM EDTA, 50 μM DTT, 5% glycerol and herring sperm DNA (25 μg ml$^{-1}$). After incubation at 37˚C for 20 min, samples were separated on 6% polyacrylamide gels containing 2% glycerol. Gels were analysed using a Bio-Rad Molecular Imager FX and Quantity One software (Bio-Rad).

## CFH binding ELISA

To investigate CFH binding by ELISA, 96-well plates (F96 MaxiSorp; Nunc) were coated with recombinant fHbp (5 μg/well) overnight at 4˚C prior to blocking with 3% bovine serum

albumin (BSA) in PBS with 0.05% Tween 20 at 37˚C. Plates were incubated with dilutions of CFH (Sigma). Binding was detected with anti-CFH mAb (OX24) and HRP-conjugated goat anti-mouse polyclonal antibody (Dako), and visualized with 3,3′,5,5′-tetramethylbenzidine (TMB) substrate reagent (Roche) and 2 N sulphuric acid stop solution (Roche) according to the manufacturer's instructions, and the $A_{450}$ measured (SpectraMax M5; Molecular Devices).

### Serum assays

Pooled normal human serum (NHS) were used in serum assays, and heat inactivated (NHS-HI) by heating at 56˚C for 30 min. *N. meningitidis* was grown overnight on Brain Heart Infusion (BHI) agar, and then $10^4$ CFU were incubated in dilutions of NHS or NHS-HI for 30 min at 37˚C in the presence of $CO_2$. Bacterial survival was determined by plating onto BHI agar in triplicate. Percent survival was calculated by comparing bacterial recovery in serum with recovery from samples containing no serum. Significance was analysed by two-way ANOVA (GraphPad Prism v.8.0).

### Flow cytometry

*N. meningitidis* was grown on BHI agar at 37˚C prior to fixation for two hours in 3% paraformaldehyde. Surface localisation of fHbp was detected using anti-fHbp pAbs and goat anti-mouse IgG-Alexa Fluor 647 conjugate (Molecular Probes, LifeTechnologies). Samples were run on a FACSCalibur (BD Biosciences), and at least $10^4$ events recorded before results were analysed by calculating the geometric mean fluorescence intensity in FlowJo vX software (Tree Star).

### SHAPE RNA secondary structure analysis

SHAPE experiments were performed using RNA transcribed *in vitro* from cDNA sequence [85]. The DNA templates contained a double-stranded T7 RNA polymerase promoter sequence (TTCTAATACGACTCACTATA) followed by the sequence of interest (S4 Table). RNA purification was done with an RNA clean kit (Zymo research); RNA concentrations were measured on a Nanodrop 100 spectrophotometer. RNA chemical modification was performed in volumes of 30µl with 1.5pmol of RNA within Folding buffer (50 mM HEPES pH 8.0, 16.5 mM $MgCl_2$). RNA samples were pre-heated at 65˚C for 3 mins and immediately incubated at 30˚C, 37˚C or 42˚C water baths for 30 mins. The modification reagent N-methylisatoic anhydride (NMIA) was added at increasing concentrations between 0 and 13mM, with DMSO (no NMIA) as control. Modification reactions were incubated for another 45mins before ethanol precipitation [86,87]. Reverse transcription was performed using Super Script III reverse transcriptase (Invitrogen). $^{32}$P-labeled reverse transcription primers (GV1-3) are listed in S4 Table. Electrophoresis on 8% (vol/vol) polyacrylamide gels was then performed to separate fragments. Band-intensities were quantified using SAFA, version 1.1 Semi-Automated Footprinting Analysis [88].

All structure calculations were performed using RNAstructure software [89]. ΔG˚SHAPE free energy change values were added to the thermodynamic free energy parameters for each nucleotide [90,91]. Pseudoknot-energy parameters were used in calculation of ΔG˚(SHAPE), according to the equation, $\Delta G˚SHAPE(i) = m\ ln[SHAPE\ reactivity(i) + 1] + b$. In this analysis, parameters were optimized at m = 0.3 and b = -1.2 kcal/mol for *fHbp*$_{s-7}$T/$_{s13}$A; m = 0.4 and b = -2.0 kcal/mol for *fHbp*$_{s-7}$T/$_{s13}$G; nucleotides with normalized SHAPE reactivities 0–0.40, 0.40–0.85, and >0.85 correspond to low, medium, and highly reactive positions, respectively[60,61]. Secondary structures were rendered using VARNA [92].

## Supporting information

**S1 Fig. The 20 most significant principal components (PCs) by a Wald test, testing for association between PCs and case-control status.** A Bonferroni correction was applied to the significance threshold for the number of non-redundant PCs.
(PNG)

**S2 Fig. Alignments of the 82 genomes containing the MC58 (NC_003112.2) serogroup B *csb* gene in the discovery sample collection aligned with the MC58 *csb* gene positions 76940–76830 on the reverse strand, the coding strand for *csb* with all A alleles coloured.** Each row shows an alignment of a contig (C) with the reference (R) from BLAST (Camacho et al. 2009). On the left, red indicates that the isolate was sampled from a patient with invasive disease, grey from a carrier. In the alignments, grey indicates identity between the contig and the reference. Polymorphisms are coloured by the alleles (A = green; C = blue; G = black; T = Orange) plus all invariant positions with the A allele are coloured green. Insertions and deletions are shown in white. The top line shows the bases of the MC58 *csb* gene in the region. Vertical lines show the region where the 21 significant *csb* kmers mapped. Sample names are shown as BIGS IDs from pubMLST and sample names shown in bold contained the 21 significant kmers.
(PNG)

**S3 Fig. Close up of the significant kmers aligned to the intergenic region between *ctrE* and *ctrF* in the discovery sample collection.** The reference genome FAM18 is shown at the bottom of the figure, grey for invariant sites and coloured at variant site positions. The kmers which map to the region shown are then plotted from least significant at the bottom to most significant at the top. The black dashed line indicates the Bonferroni-corrected significance threshold–all kmers above the line are significantly associated with the phenotype. The background colour of the kmers represents the direction of the association, grey when $\beta < 0$ (carriage-associated) and dark orange when $\beta > 0$ (disease-associated). Kmers are coloured by their allele at all variant positions (A = green; C = blue; G = black; T = Orange). The *E. coli* consensus for the -10 and -35 promoter regions are shown above the kmers aligned with the major and minor alleles in the discovery sample collection at these positions. Matches to the consensus are shown in bold.
(PNG)

**S4 Fig. Close up of the significant kmers aligned to the gene *tspB* in the discovery sample collection.** The reference genome FAM18 is shown at the bottom of the figure, grey for invariant sites and coloured at variant site positions. The kmers which map to the region shown are then plotted from least significant at the bottom to most significant at the top. The black dashed line indicates the Bonferroni-corrected significance threshold–all kmers above the line are significantly associated with the phenotype. The background colour of the kmers represents the direction of the association, grey when $\beta < 0$ (carriage-associated) and dark orange when $\beta > 0$ (disease-associated). Kmers are coloured by their allele at all variant positions (A = green; C = blue; G = black; T = Orange).
(PNG)

**S5 Fig. Genes or intergenic regions containing only significant kmers associated with carriage *versus* invasive disease in 261 isolates sampled from the Czech Republic in 1993.** Each plot shows the midpoint of the kmer association within the gene/intergenic region +/- 10kb. Open circles represent a SNP aligned to the reference genome FAM18, a cross represents the left-most mapping position of a kmer in the reference genome FAM18 based on mapping and

BLAST alignments. Significant kmers are coloured by the LMM estimated direction of effect. Bonferroni-corrected significance thresholds are shown by black dashed (SNPs) and dotted (kmers) lines. Gene names separated by colons indicate intergenic regions. FAM18 reference genome gene name prefixes have been shortened from NMC_RS to NMC_.
(PNG)

**S6 Fig. Close up of the significant kmers aligned to the gene *fba* in the discovery sample collection.** The reference genome FAM18 is shown at the bottom of the figure, grey for invariant sites and coloured at variant site positions. The kmers which map to the region shown are then plotted from least significant at the bottom to most significant at the top. The black dashed line indicates the Bonferroni-corrected significance threshold–all kmers above the line are significantly associated with the phenotype. The background colour of the kmers represents the direction of the association, grey when $\beta < 0$ (carriage-associated) and dark orange when $\beta > 0$ (disease-associated). Kmers are coloured by their allele at all variant positions (A = green; C = blue; G = black; T = Orange). The *E. coli* consensus for the FNR box DNA binding site and the *fHbp* promoter FNR binding site are shown above the kmers aligned with the major and minor alleles in the discovery sample collection at these positions.
(PNG)

**S7 Fig. UPGMA tree of 1,295 ST-41/44 complex *N. meningitidis* genomes downloaded from PubMLST.org.** The UPGMA tree, used solely for visualisation, was estimated using a distance matrix calculated from the kmer presence/absence matrix. The most common serogroups are shown on the outer ring. Disease status is shown on the next ring, invasive disease (red, n = 1,046) or carriage (grey, n = 249). Presence of the two significant kmers in *fba* in the discovery sample collection are shown in black in the inner ring.
(PNG)

**S8 Fig.** (A) The 126 and 249 bp fragments used for EMSA. (B) Sequences upstream of *fHbp* were amplified and digested with *Hae*III, end labelled with [γ-$^{32}$P]-ATP, then incubated in increasing concentrations of FNR (0, 0.75, 1.5, and 3 μM).
(PNG)

**S9 Fig. Close-up of the ST-41/44 complex replication analysis kmers aligned to the start codon of *fHbp* and the surrounding region.** The reference genome FAM18 is shown at the bottom of the figure, grey for invariant sites and coloured at variant site positions. The kmers above 1% minor allele frequency that map or align to the region shown are plotted from least significant at the bottom to most significant at the top. The background colour of the kmers represents the direction of the association, grey when $\beta < 0$ (carriage-associated) and dark orange when $\beta > 0$ (disease-associated). Kmers are coloured by their allele at all variant positions (A = green; C = blue; G = black; T = Orange). The *fHbp* start codon is shown aligned above the kmers in red, the ribosome binding site (RBS) in blue and two putative anti-RBSs in orange. The most significant kmers plus the SNPs tested experimentally are labelled.
(PNG)

**S10 Fig. Close up of the ST-41/44 complex replication analysis kmers aligned to the *fHbp* stop codon and upstream region.** The reference genome FAM18 is shown at the bottom of the figure, grey for invariant sites and coloured at variant site positions. The kmers which map to the region shown are then plotted from least significant at the bottom to most significant at the top. The background colour of the kmers represents the direction of the association, grey when $\beta < 0$ (carriage-associated) and dark orange when $\beta > 0$ (disease-associated). Kmers are coloured by their allele at all variant positions (A = green; C = blue; G = black; T = Orange).

The *fHbp* stop codon is annotated above the aligned kmers. The most significant kmer plus the SNP tested experimentally are labelled.
(PNG)

**S11 Fig. Close up of the ST-41/44 complex replication analysis kmers aligned to the *fHbp* positions 643–760.** The reference genome FAM18 is shown at the bottom of the figure, grey for invariant sites and coloured at variant site positions. The kmers which map to the region shown are then plotted from least significant at the bottom to most significant at the top. The background colour of the kmers represents the direction of the association, grey when $\beta < 0$ (carriage-associated) and dark orange when $\beta > 0$ (disease-associated). Kmers are coloured by their allele at all variant positions (A = green; C = blue; G = black; T = Orange). The *fHbp* stop codon is annotated above the aligned kmers in red.
(PNG)

**S12 Fig. We tested *fHbp* Novartis variants from pubMLST for their association with carriage *vs*. IMD by LMM, using the full kmer kinship matrix to control for population structure.** Correcting the significance threshold for the number of variants, variant 2.24 was significantly associated with disease status. The codon each variant contains at codon 261 (relative to the FAM18 reference fHbp) is shown at the top. Colour represents the β point estimate, the direction of the effect of the association by the LMM.
(PNG)

**S13 Fig. Secondary structure of *fHbp*$_{s\text{-}7}$T/$_{s13}$A RNA calculated using RNA structure based on SHAPE reactivity data at (A) 30˚C and (B) 42˚C.** SHAPE reactivity data are mapped on the RNA structure and colour coded by intensity as shown on the bars; the RBS is circled in red. (C) NMIA modifications for reactions conducted at 30˚C (Lane1- 3); 37˚C (lane 4–6) and 42˚C (lane 7–9), with a gradient of 0-13mM NMIA, analysed by denaturing polyacrylamide gel electrophoresis. (D) SHAPE reactivity profile at different temperatures; nucleotides with temperature dependent changes in reactivity are listed in the table as strong (++), medium (+) and weak (-).
(PNG)

**S14 Fig.** Secondary structure of the *fHbp*$_{s\text{-}7}$C/$_{s13}$G RNA calculated using RNA structure based on SHAPE reactivity data at **(A)** 30˚C and **(B)** 42˚C; SHAPE reactivity data are mapped on the RNA structure and colour coded by intensity as shown on the bars; the RBS is circled in red. **(C)** NMIA modification for reactions conducted at 30˚C (Lane 1–3); 37˚C (lane 4–6) and 42˚C (lane 7–9), with a gradient of 0-13mM NMIA, analysed by denaturing polyacrylamide gel electrophoresis. **(D)** SHAPE reactivity profile at the different temperatures; nucleotides with temperature dependent changes in reactivity are listed in the table as strong (++), medium (+) and weak (-).
(PNG)

**S15 Fig. Serum sensitivity assays of *N. meningitidis* strains using heat-inactivated serum.** SNPs; *fHbp*$_{s\text{-}7}$T/*fHbp*$_{s13}$G (blue), *fHbp*$_{s\text{-}7}$T/*fHbp*$_{s13}$A (green), *fHbp*$_{s\text{-}7}$C/*fHbp*$_{s13}$G (red) and *fHbp*$_{s\text{-}7}$C/*fHbp*$_{s13}$A (black) demonstrated no statistical difference in bacterial survival. Error bars show SD (n = 3) with statistical analysis performed in Prism v10 using One-way ANOVA.
(PNG)

**S1 Table. Strains used to test *fba* and *fHbp* SNPs.**
(PDF)

**S2 Table. Plasmids.**
(PDF)

**S3 Table. Oligonucleotide Primers.**
(PDF)

**S4 Table. Sequences used for SHAPE analysis.**
(PDF)

**S5 Table. Comparison of the alleles tested by Spinsanti *et al*. to those used to test *fHbp*$_{s-7}$C/
T and *fHbp*$_{s13}$G/A.**
(PDF)

**S6 Table. Czech isolates used in the study.**
(CSV)

**S7 Table. c41/44 isolates used in the study.**
(CSV)

## Acknowledgments

This work made use of the PubMLST.org website developed by Martin Maiden and Keith Jolley and located in the University of Oxford.

## Author Contributions

**Conceptualization:** Sarah G. Earle, Rachel M. Exley, Gabriele Varani, Christoph M. Tang, Daniel J. Wilson, Martin C. J. Maiden.

**Data curation:** Sarah G. Earle, Vasiliki Kostiou, Odile B. Harrison, Holly B. Bratcher, Martin C. J. Maiden.

**Formal analysis:** Sarah G. Earle, Mariya Lobanovska, Hayley Lavender, Changyan Tang, Rachel M. Exley, Elisa Ramos-Sevillano, Douglas F. Browning, Daniel J. Wilson.

**Funding acquisition:** Gabriele Varani, Christoph M. Tang, Daniel J. Wilson, Martin C. J. Maiden.

**Investigation:** Sarah G. Earle, Mariya Lobanovska, Hayley Lavender, Changyan Tang, Elisa Ramos-Sevillano, Douglas F. Browning, Odile B. Harrison, Holly B. Bratcher, Daniel J. Wilson, Martin C. J. Maiden.

**Methodology:** Sarah G. Earle, Changyan Tang, Daniel J. Wilson, Martin C. J. Maiden.

**Project administration:** Sarah G. Earle, Rachel M. Exley, Christoph M. Tang, Daniel J. Wilson, Martin C. J. Maiden.

**Resources:** Sarah G. Earle, Mariya Lobanovska, Hayley Lavender, Vasiliki Kostiou, Odile B. Harrison, Holly B. Bratcher, Daniel J. Wilson, Martin C. J. Maiden.

**Software:** Sarah G. Earle, Vasiliki Kostiou, Daniel J. Wilson.

**Supervision:** Gabriele Varani, Christoph M. Tang, Daniel J. Wilson, Martin C. J. Maiden.

**Validation:** Sarah G. Earle, Mariya Lobanovska, Hayley Lavender, Changyan Tang, Odile B. Harrison, Holly B. Bratcher, Christoph M. Tang, Daniel J. Wilson, Martin C. J. Maiden.

**Visualization:** Sarah G. Earle, Changyan Tang, Daniel J. Wilson.

**Writing – original draft:** Sarah G. Earle, Mariya Lobanovska, Hayley Lavender, Changyan Tang, Gabriele Varani, Christoph M. Tang, Daniel J. Wilson.

**Writing – review & editing:** Sarah G. Earle, Mariya Lobanovska, Hayley Lavender, Gabriele Varani, Christoph M. Tang, Daniel J. Wilson, Martin C. J. Maiden.

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
