## [Decision Letter · Decision Letter 0]

24 Jun 2021

Dear Prof. Maiden,

Thank you very much for submitting your manuscript "Genome-wide association studies reveal the role of polymorphisms affecting factor H binding protein expression in host invasion by Neisseria meningitidis" for consideration at PLOS Pathogens. As with all papers reviewed by the journal, your manuscript was reviewed by members of the editorial board and by several independent reviewers. Based on the reviews, we are likely to accept this manuscript for publication, providing that you modify the manuscript according to the review recommendations.

Sincerely,

Xavier Nassif

Section Editor

PLOS Pathogens

Xavier Nassif

Section Editor

PLOS Pathogens

Kasturi Haldar

Editor-in-Chief

PLOS Pathogens

orcid.org/0000-0001-5065-158X

Michael Malim

Editor-in-Chief

PLOS Pathogens

orcid.org/0000-0002-7699-2064

Reviewer Comments (if any, and for reference):

Reviewer's Responses to Questions

**Part I - Summary**

Reviewer #1: This manuscript uses genome-wide association studies of a large collection of Neisseria meningitides (Mc) isolates to search for genomic difference between invasive and carriage Mc strains. They identify several polymorphisms that would alter the expression of the factor H binding protein (FHBB). They found association within a few capsule genes, the MDA phage that was previously associated with invasive disease, and several other loci. This manuscript follows that of Spinsanti et al (PloS pathogens published in March 2021) that focused only of fHBP to identify a similar finding about expression and the same mechanistic basis. The authors argue that since their study was finished before the Spinsanti study was published and that they used different methods this should not preclude the publication of this work. I am supportive of this viewpoint since the studies are complementary but I believe the manuscript could be improved.

Reviewer #2: This work by Sarah G . Earle et al is focused on bacterial genome-wide association studies of N. meningitidis. The events that lead to meningococcal invasion are still elusive. Here, SGE et al explored the relationship between genetic variation and invasive meningococcal disease in GWAS. In parallel, an independent study was published in PloS Pathogen and dealing with the same set of publicly available data (Spinsanti et al). In contrast to this latter work, SGE et al analysed the data set thoroughly and extracted several interesting associations. Among these, the authors chose to focus on fHbp.

This clear and concise manuscript highlights a very important topic and goes much further than Spinsanti et al in terms of GWAS analysis. This in-depth analysis provided interesting results and will be an important addition to the community. This justifies publication in PloS Pathogen.

**Part II – Major Issues: Key Experiments Required for Acceptance**

Reviewer #1: None

Reviewer #2: The experiments on fHbp are well documented, although it is to be regretted that the authors did not address other SNPs and Kmers. Considering the early online publication of Spinsanti et al, I would not suggest further experiments. However, and considering the very short discussion, I would really appreciate that the authors discuss other GWAS association, especially those presented in Fig1:

- fba is an important finding. This gene was found to be essential in the blood of mice. However, the authors were only interested in fHbp. Modification of fba sequence may alter its metabolic function.

- pilV is an interesting finding since it has been proposed as an adhesin.

- mafB was described as par of a toxin-antiToxin system.

- lptF is involved in LOS maturation

- some genes are involved in RNA an tRNA maturation (gidA, vacB)

- trkH may regulate K+ uptake, which may be related to resistance against the host.

**Part III – Minor Issues: Editorial and Data Presentation Modifications**

Reviewer #1: 1. The Discussion is very short and does not really compare and contrast between the two studies. This needs to be discussed and they need to go into detail.

2. They mention two other polymorphisms in the Results and show others in Figure 1 but do not mention the others in the Results nor any of these in the Discussion. The strength of this whole genome study relative to the focused Spinsanti paper is the identification of these other loci. They each should be given some attention in the Results and the Discussion.

Reviewer #2: -

PLOS authors have the option to publish the peer review history of their article (what does this mean?). If published, this will include your full peer review and any attached files.

Reviewer #1: **Yes: **H Steven Seifert

Reviewer #2: No

Figure Files:

Data Requirements:

Reproducibility:

References:

---

## [Editor Report · Decision Letter 1]

29 Sep 2021

Dear Prof. Maiden,

We are pleased to inform you that your manuscript 'Genome-wide association studies reveal the role of polymorphisms affecting factor H binding protein expression in host invasion by Neisseria meningitidis' has been provisionally accepted for publication in PLOS Pathogens.

Best regards,

Xavier Nassif

Section Editor

PLOS Pathogens

Xavier Nassif

Section Editor

PLOS Pathogens

Kasturi Haldar

Editor-in-Chief

PLOS Pathogens

orcid.org/0000-0001-5065-158X

Michael Malim

Editor-in-Chief

PLOS Pathogens

orcid.org/0000-0002-7699-2064
---

## [Editor Report · Acceptance letter]

13 Oct 2021

Dear Prof. Maiden,

We are delighted to inform you that your manuscript, "Genome-wide association studies reveal the role of polymorphisms affecting factor H binding protein expression in host invasion by Neisseria meningitidis," has been formally accepted for publication in PLOS Pathogens.

Best regards,

Kasturi Haldar

Editor-in-Chief

PLOS Pathogens

orcid.org/0000-0001-5065-158X

Michael Malim

Editor-in-Chief

PLOS Pathogens

orcid.org/0000-0002-7699-2064